# Cross-Cultural Information for Japanese Nurses at an International Hospital: A Controlled Before–After Intervention Study

**DOI:** 10.3390/ijerph191912829

**Published:** 2022-10-07

**Authors:** Mariko Nishikawa, Masaaki Yamanaka, Akira Shibanuma, Junko Kiriya, Masamine Jimba

**Affiliations:** 1Department of Global Health and Nursing, Graduate School of Nursing, University of Human Environments, Obu 474-0035, Japan; 2Department of Maritime Science and Technology, Japan Coast Guard Academy, Kure 737-0832, Japan; 3Department of Community and Global Health, Graduate School of Medicine, The University of Tokyo, Tokyo 113-8654, Japan

**Keywords:** digital video, cross-cultural information, foreign patient, nurse, Japan

## Abstract

This study sought to evaluate the efficacy of providing health information through an ordinary travel guidebook combined with a short digital video compared with an ordinary travel guidebook alone by measuring the anxiety levels of Japanese nurses dealing with foreign patients. We conducted a controlled before–after intervention study in 2016 at a major international hospital in Japan. We created two interventions: (1) a brief piece of health information from a travel guidebook for Japan, (2) the same travel guidebook, and a four-minute digital video in English on health information in Japan, titled *Mari Info* Japan for nurses. After each intervention, we assessed the nurses’ levels of anxiety about caring for foreign patients. We evaluated the results through statistical testing and the State–Trait Anxiety Inventory Form Y. Of 111 nurses, 83 (74.8%) completed both interventions and the questionnaires. The second intervention (the guidebook and video) proved more effective than the first (the guidebook) for reducing anxiety related to caring for foreign patients. Japanese nurses can lower their anxiety about dealing with foreign patients by learning about the content of various forms of health care information currently accessible to overseas visitors. Using both guidebooks and digital videos can help to reduce nurses’ anxiety.

## 1. Introduction

The number of overseas visitors increased by 21.4%, from 24 million in 2016 to over 31.8 million in 2019, as a result of a successful government strategy [1,2]. In 2019, the government of Japan passed legislation permitting longer temporary work permits for both skilled and unskilled foreign workers to compensate for a decline in the working-age population [1,3]. A concomitant increase in overseas patients to Japanese health facilities could be expected with an increase in visitors and expatriates. Common ailments among foreigners who visit health facilities include fever, bruises, nausea, abdominal pain, and diarrhea [4,5,6]. In extreme cases, they require hospital stays and operations [7,8].

Since nurses work with patients more closely than other medical professionals, as is the nature of their profession, cross-cultural concerns and foreign patients’ wishes are very important to both nurses and foreign patients [9]. The provision of quality care by nurses to foreign patients is impacted by the language of communication, payment, and cross-cultural information [10,11,12]. Providing cross-cultural information to health care workers helps them make foreign patients feel more at ease [9,13,14]. Internationally, most nurses are unlikely to have cross-cultural education, information, or training. For example, this has been observed in Switzerland, a developed country similar to Japan, where many multinational people work together [15].

A review of the literature [16,17,18] identified three critical concerns for nurses. Language skills training in intercultural health care communication is the first concern. Nurses are not confident in their cross-cultural skills in their second language. For example, nurses in Austria needed special training to develop language and cultural skills [16]. Providing cross-cultural care information for nurses is the second concern. For example, culturally and linguistically appropriate guidelines were useful in shortening the gap between Arabic and Western cultures to meet the health care needs of the Arabic population in the United States [17]. Cross-cultural education for nursing students is the third concern. For example, some education programs in the United States include a study abroad program in a developing country to understand customs, living conditions, and health institutions. Those who participated in these programs reported improved cross-cultural skills in their professions [18].

The standard of Japanese nursing requires the provision of competent care to all patients regardless of age, ethnicity, language, or nationality [19]. However, Japan is a mono-ethnic and monolingual nation, making it difficult to understand the backgrounds of overseas visitors. Nurses in Japan deal with three main areas of concern while taking care of overseas patients: culturally sensitive care, medical expenses, and the language of communication, including informed consent [20]. Sufficient cultural education could reduce nurses’ concerns and lead to culturally sensitive care [17,18].

Japan has been encouraging classes and short study abroad programs that immerse participants in different cultures in the university nursing curriculum for a decade. However, these programs have only reached a few nursing students. The health sections of standard travel guidebooks for overseas visitors to Japan provide only brief messages and do not contain cultural information for these visitors. Therefore, we need more new and interesting ways to effectively provide Japanese health information to nurses.

A digital video is a simple way to educate patients with disabilities, such as those that have dementia [21]. A previous study showed that an educational digital video of washing hands with soap in India improved the sanitation level of multiple households [22]. Internationally, other examples of patient education and health care provider training through digital videos have included the improvement of health literacy for self-care among older Australians [23] and insulin therapy among individuals with diabetes in England [24].

We conducted a cross-sectional survey of nurses [19] and another survey of foreign visitors to Japan [25] prior to this study because we could not find a culturally relevant, easy-to-watch video related to health care in Japan. Those nurses were anxious and concerned about caring for foreigners due to a lack of knowledge of their cultural background, not just their language [19]. We analyzed the results and produced a digital video named *Mari Info Japan* with information pertinent to accessing medical services in Japan. The video helped nurses to better understand how to interact with people from different countries.

The objective of our study was to investigate the effectiveness of reducing anxiety among Japanese nurses who care for foreign patients in Japan. Our hypothesis was that Japanese nurses could reduce their level of anxiety in dealing with foreign patients by reading an ordinary travel guidebook and watching the digital video named *Mari Info Japan* on cross-cultural and health system information in Japan. The effectiveness of providing cross-cultural information through an ordinary guidebook and a digital video to nurses has not been explored.

## 2. Methods

### 2.1. Setting

This study was performed at an international hospital in the city of Kobe, Japan. We chose this hospital because of its history and location. Kobe is an international seaport with a history of foreign settlement. The hospital was established by a German doctor in 1871 [26]. The hospital has since treated seamen, diplomats, and migrants from the Far East and Western countries for decades. Today, the hospital has a unique international internal medicine outpatient unit staffed by medical doctors and nurses fluent in both Japanese and English. Approximately 200 foreign patients visited the hospital each month in 2014. Most of them were English speakers. Of the foreign patients, 70% were migrants to Japan, while the remaining 30% were short-term overseas visitors.

Nurses who work at the hospital take care of foreign patients on a daily basis. Two-thirds of the Japanese nurses (120 out of 180) working at the hospital at the time of the study could read and correspond in English. Some spoke another foreign language, such as French or Mandarin. Many nurses wore a badge depicting the country flags of the languages they could speak. All nurses were born in Japan and were of Japanese nationality. We chose this hospital because 120 of the nurses clearly understood our research interventions in English, and all 180 nurses regularly cared for foreign patients.

#### 2.1.1. Intervention Materials

(1) A digital video: We created a digital video movie called *Mari Info Japan* to inform overseas visitors about the health system in Japan [25,27]. The digital video was based on our two previous cross-sectional studies involving 1343 overseas visitors at Narita International Airport and 513 nurses at 320 hospitals in Japan [19]. We identified their requests for content and information regarding the main concerns of foreign patients [12]. The video provided information to overseas visitors on the culture and health system in Japan. The digital video comprised 11 points succinctly presented with music [25].

(2) A travel guidebook: The guidebook contained one paragraph with brief information about health insurance in Japan and was sold among overseas visitors internationally [28].

#### 2.1.2. Study Design and Procedures

We performed a controlled before–after intervention study [29] involving two interventions: (1) an ordinary travel guidebook that is popular with overseas visitors and (2) the same travel guidebook and a digital video. In this controlled before–after intervention study, we evaluated the anxiety levels of nurses at the selected international hospital from January to March 2016. The nurses responded to our questions about their levels of anxiety before receiving either intervention. After each intervention, the nurses responded to the same questions to assess their anxiety level in dealing with foreign patients. We examined the difference in anxiety levels pre- and post-intervention and between the first intervention (the guidebook) and the second intervention (the guidebook and video). The interventions were conducted within one month (Figure 1).

### 2.2. Participant Entry

#### 2.2.1. Eligibility

We contacted Japanese nurses aged 20 years and older who could read and understand English in three out of six wards. Their ability to comprehend English was determined by the head nurse and the nurses themselves.

#### 2.2.2. Enrollment Procedure

Subsequently, we enlisted all eligible nurses who understood English in the three wards. It was difficult to have a control group that could avoid being exposed to the interventions at this small international hospital. We first sampled nurses for the guidebook alone group. Within one month, the same nurses from the same wards participated in the digital video group.

### 2.3. Interventions

We provided a short explanation of how to use the digital video to nursing committees, after which the nurses answered the questionnaire without any assistance or interference.

#### 2.3.1. First Intervention Procedure

The nurses who participated in our research read a brief piece of information in English from an ordinary travel guidebook [28].

#### 2.3.2. Second Intervention Procedure

Within one month, the same nurses viewed a four-minute digital video called *Mari Info Japan* in English for the 2nd intervention. The nurses used an 8 × 5 inch digital notepad with headphones to watch the video.

### 2.4. Outcome Assessment

Pre- and post-intervention, we examined the level of anxiety with the *Mari Meter-X* and State–Trait Anxiety Inventory Form Y (STAI-Y).

1. The *Mari Meter-X* [20,25] was used to analyze the culture-related anxiety levels of nurses when caring for foreign patients in Japan. This 15-item questionnaire uses a five-point Likert-type scale. The total score ranges from 15 to 75 points. A higher score indicates greater anxiety.

2. The STAI-Y [30] is a questionnaire that uses a four-point Likert-type scale and consists of 40 items. The questions are separated into two parts. The first half consists of questions about respondents’ state anxiety, and the second half consists of questions about their trait anxiety. State anxiety referred to the level of anxiety nurses felt when taking care of foreign patients, and the level of anxiety they experienced in their daily routine was described by their trait anxiety. Before the intervention, the nurses answered questions regarding both state and trait anxiety. After the first and second interventions, the nurses reported only state anxiety. The STAI-Y scores (ranging from 40 to 160) were divided into state anxiety (ranging from 20 to 80 points) and trait anxiety (ranging from 20 to 80 points). In both cases, a higher score implied greater anxiety.

#### 2.4.1. Other Information

As this was a controlled before–after intervention study, we gathered demographic data. This included nurses’ age, gender, profession, educational level, work experience, current position, and ward of work (Table 1). The nurses took approximately 15 min to answer all questions. Nurses could choose a convenient time to participate in this study. All the nurses had completed classes in English as a second language since junior high school; however, none of them had completed cross-cultural competency care classes or a study abroad program before this study.

#### 2.4.2. Bias Prevention

We considered three biases carefully. The first was selection bias. The three wards were selected randomly by the head of the hospital, who was not involved in our research. The second was performance bias. The nurses were not told whether they were in the first or second intervention session. The third was attrition bias. We analyzed only nurses who completed both interventions. Intention-to-treat analysis was not performed in this study because the denominators of both groups were the same (*n* = 83).

#### 2.4.3. Data Analysis

For the primary outcome, we selected a nonparametric statistic because Levene’s test was statistically significant (*p* < 0.05) in the nonpaired groups. We examined the difference in median scores for the *Mari Meter-X* and STAI-Y state anxiety before and after the intervention for each group using a two-sided Wilcoxon signed rank test. When conducting the Wilcoxon rank sum test, we analyzed the differences in the calculated scores between groups and between pre- and post-intervention.

To achieve confidence levels of 95% and 80% and detect a difference of 0.2, we deter-mined that a sample size of 80 nurses was necessary for each intervention. The rationale was based on previous research examining the effects of video guidance on anxiety reduction [25]. We calculated Cronbach’s alpha internal consistency reliability coefficient to assess the reliability of the measures used in this study. The Mari Meter-X yielded a reliability of 0.88, while the STAI-Y yielded a reliability of 0.91. All statistical evaluations were performed with the help of the JMP statistical package (version 14.0, SAS Campus Drive Cary, NC 27513, USA).

## 3. Results

A total of 111 nurses agreed to participate in the study. We eliminated data from 28 nurses who did not complete an intervention or the questionnaire because they became unavailable during the intervention (13 in the first intervention and 15 in the second intervention) or because their questionnaires contained incomplete data (Figure 1). We collected and analyzed data from 83 nurses who completed both interventions. The nurses were the same in both interventions to ensure consistency across the study.

### 3.1. Characteristics of Nurses

An overview of the basic features of the participating nurses (*n* = 83) is provided in Table 1. The mode of age fell within the 30–40 year bracket. Most of the nurses were female (92.8%) and held a registered nurse (RN) qualification (97.6%). Over half of the nurses had a three-year RN diploma (62.7%). None of them had taken cross-cultural nursing courses prior to the study. The average duration of working experience was 12.8 years. Four out of five nurses worked as staff members. Most of the nurses worked in an inpatient unit when they participated in this study.

### 3.2. Change in Levels of Anxiety

Table 2 shows both the general feeling the nurses had before the interventions and the change induced by the interventions. The former is indicated by the 25th (41), 50th (46), and 75th (53) percentiles out of the maximum 80 points in the STAI-Y trait anxiety scores. The change in anxiety level is evident between the pre- and post-intervention. In the first intervention group, the *Mari Meter-X* score did not change significantly (*p* = 0.151) after the intervention, whereas the STAI-Y state anxiety score changed significantly (*p* < 0.001).

A statistically significant reduction in the *Mari Meter-X* scores from before to after the intervention was observed in the second intervention group (*p* < 0.001). Similarly, the reduction in the STAI-Y state anxiety score from before to after receiving the second intervention was statistically significant (*p* < 0.001). A statistically significant reduction in the *Mari Meter-X* scores from before to after treatment was observed in the combined first and second intervention groups (*p* < 0.001). However, there was a statistically significant increase in the STAI-Y state anxiety scores from before to after receiving the first and second interventions (*p* < 0.01).

Table 3 shows the comparison of the change in scores between the two groups using the Wilcoxon rank-sum test. The *Mari Meter-X* score was significantly reduced in the second intervention group compared to that in the first intervention group (*p* < 0.005). The STAI-Y state anxiety was not significantly changed in the first intervention group and second intervention group (*p* = 0.088).

Table 4 indicates the contrast in the shift in scores between the two groups using the Wilcoxon rank-sum test. The *Mari Meter-X* score was significantly decreased in the combined first and second intervention groups compared with that in the first intervention group (*p* < 0.001). However, the STAI-Y state anxiety scores were significantly different in the first intervention group and the first and second intervention groups (*p* = 0.003).

## 4. Discussion

This study showed that the combination of an ordinary guidebook and digital video reduced concerns related to caring for foreign patients among Japanese nurses who had not been on a study abroad program or completed a cross-cultural competency class. No nurses had completed a cultural competency course, even though the competency standard of Japanese nursing education requires treating foreign patients the same as Japanese people.

Japanese nurses can reduce their anxiety about taking care of foreign patients by learning about the various forms of health care information that is currently available to overseas visitors in both guidebook and digital form. The appropriate amount of information helps them gain knowledge of the cross-cultural norms of the foreign patients they might care for [20] and can therefore enable culturally sensitive care [16,18]. The information on the Japanese health system and illness prevention was related to ethical concerns and cultural norms (measured with *Mari Meter-X*). However, the intensity of feelings of nervousness (measured with the STAI-Y) was not found in the nurses.

Our results were similar to those of a previous study that confirmed the advantages of providing succinct cross-cultural health information to international visitors to Japan [25]. Similar efforts were made to use digital learning resources at a Norwegian university. The study examined the usability and value of learning of e-compendiums shared and implemented across three European universities, including two invited universities in Spain and the UK [31]. People with dementia may have health needs but have limited language capacity. In this situation, a digital video that provided guidance was found to be the most helpful [21]. Similarly, an educational DVD was found to reduce anxiety in women diagnosed with gestational diabetes mellitus in the UK [32].

The main finding confirms that reading an ordinary guidebook and watching a digital video provides nurses with knowledge of what foreign visitors to the country know about the Japanese health system. None of the nurses who worked at the international hospital had taken any cross-cultural nursing courses. However, nurses’ anxiety levels improved after they were made aware of the information available to international patients in a combination of a guidebook and digital video [33]. This made the nurses more comfortable providing information on the provision of competent care for all foreign patients. Madeleine M Leininger explained that trans-cultural nursing theory emphasizes the importance of understanding how to care for patients with intercultural backgrounds [34].

It remains uncommon for nurses in a homogenous society such as Japan to learn about foreign customs. The hospital in this study had a unique history in that it was established by foreign doctors for foreign patients. The main focus is on the satisfactory treatment of patients and their health. Only limited studies and tools related to information requests from overseas visitors are available.

This research has several strengths. For the first time, this study evaluated whether nurses could reduce their anxiety related to treating foreign patients by consulting an ordinary guidebook and digital video showing the information on the Japanese health system available to foreign patients. Another strength was that we performed the study at an international hospital in Japan where the nurses cared for foreign patients daily and were proficient in English. Consequently, their opinions were more realistic than those of nurses from other hospitals. Third, this study focused on assessing interventions for addressing unknown overseas visitors’ basic requests. Last, this study had a moderately high participation rate of 93% (111 out of 120). Fourth, the advantage of using the guidebook as part of the intervention was that it showed how nurses could understand the common knowledge of overseas visitors to Japan. Fifth, there were no factors included in this study that affected the outcome because it was a controlled before–after intervention study with the same nurses.

This study also has certain limitations. First, our research design was controlled before and after the intervention study, which did not allow for another comparison group. Thus, our research results were not strong. Second, we were not certain if the nurses were blinded to their interventions, even though we tried to ensure this. Furthermore, the research assistants who directed the data collection and data analyses were not blinded to the intervention. Third, temporal changes were a major problem in our study. We could not randomly assign nurses to groups, which meant that we had only one group to evaluate the effect of the intervention on other factors that could affect the outcome. Unspecified factors remain, such as lingering effects from the first intervention and working environments. The head nurse, who was the vice president of the hospital, changed between the first and second interventions, which might have emotionally affected the nurses in our sample. However, we could not examine whether these or other factors were due to our single-group study design in a unique hospital setting.

### Implications and Future Study

Our findings demonstrate the benefits of delivering health-related information through feedback from foreign patients to nurses through a simple but comprehensive approach. Furthermore, we argue that scholars should be able to constantly assess the content of the most recent educational material that is necessary for nurses caring for foreign patients to review. Examples include web learning and the use of smartphone applications.

Future research on ways to introduce this kind of information to nurses at a convenient time for them may be important. We did not generalize the results in this study. We believe that this research setting is unique compared to other hospitals in Japan. A randomized controlled trial (RCT) is necessary to generate stronger evidence for the effectiveness of the intervention while avoiding unspecified confounding factors. We also need to measure cultural awareness of culturally competent care in other settings, such as Sweden, where approximately 20% of the population is foreign-born, and nurses are increasingly exposed to patients from diverse cultures [35] to improve patient safety and equity.

## 5. Conclusions

People are increasingly moving across different national borders, and this number is increasing rapidly. In this context, the role required of nurses working on the front line of health care is significant. Although it is inherently better for them to learn about the cultural background of their patients before going into clinical practice, they deal with foreign patients in a realistically difficult situation. In this situation, as part of their education, Japanese nurses who care for many overseas visitors but have not completed a cultural competency-care course for foreign patients were asked to learn various cross-cultural details. The results showed that gathering information from travel guidebooks as well as digital videos reduced anxiety about dealing with foreign patients.

## Figures and Tables

**Figure 1 ijerph-19-12829-f001:**
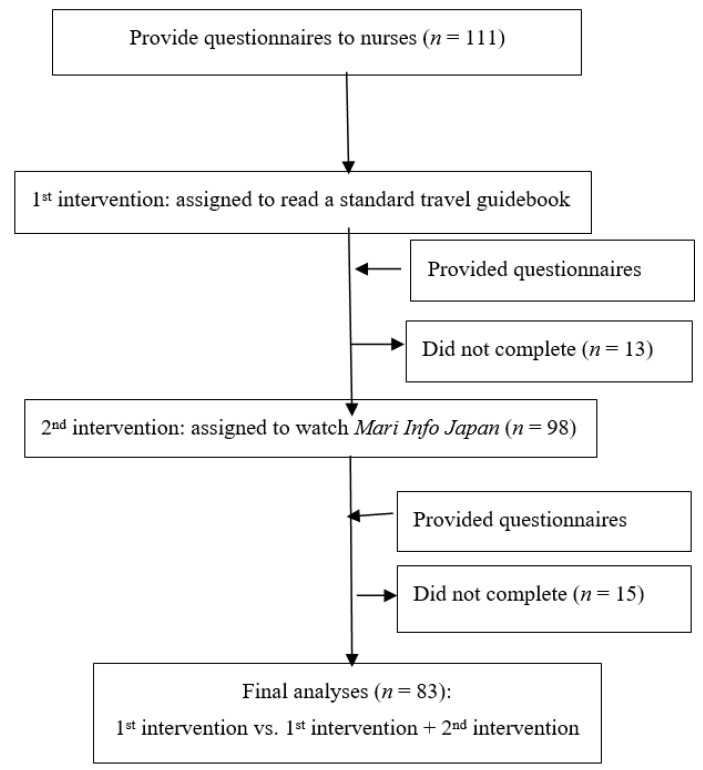
Trial profile.

**Table 1 ijerph-19-12829-t001:** Demographic profiles of nurses.

					*n* = 83
Participant Characteristics		(*n*)	%	Ave (SD)	Mode
Age					30–40
Gender	Female	(77)	92.8		
	Male	(6)	7.2		
Profession	Nurse (RN)	(81)	97.6		
	Nurse (LPN)	(2)	2.4		
Nursing education	Bachelor’s degree or more	(19)	22.9		
	Junior College	(8)	9.6		
	RN diploma (3-year)	(52)	62.7		
	LPN diploma (2-year)	(2)	2.4		
	Other	(2)	2.4		
Cross-cultural nursing course	Yes	(0)	0.0		
Work experience (year)				12.8 (8.5)	12
Current position	Unit chief nurse	(7)	8.4		
	Vice chief nurse	(10)	12.1		
	Staff nurse	(66)	79.5		
^a^ Ward of foreigner care	Inpatient unit	(61)			
	Outpatient unit/endoscopy/lab	(28)			
	^b^ ICU/Ope/ER	(3)			
	Other	(10)			
Ward of current work	Inpatient unit	(78)			
	Outpatient unit/endoscopy/lab	(4)			
	^b^ ICU/Ope/ER	(1)			

Ave = Average, SD = Standard deviation; ^a^ Multiple answers; ^b^ ICU/Ope/ER = ICU (Intensive Care Unit), Ope (Operation Room), ER (Emergency Room).

**Table 2 ijerph-19-12829-t002:** Outcomes for before vs. after the test for the same intervention groups.

									*n* = 83
	Before				After				*p* Value
		(Median)				(Median)			
	Q1	Q2	Q3	Range	Q1	Q2	Q3	Range	
STAI-Y: trait anxiety	41	46	53		-	-	-	-	-
1st intervention (Guidebook)									
*Mari Meter-x*	52	59	65	34–75	51	58	64	34–75	0.151
STAI-Y: state anxiety	41	47	52	27–67	41	45	51	27–72	0.001
2nd intervention (Digital animation)									
*Mari Meter-x*	52	59	63	36–74	46	52	61	32–75	0.001
STAI-Y: state anxiety	48	55	60	27–67	45	51	56	30–76	0.001
1st & 2nd intervention									
*Mari Meter-x*	52	59	65	34–75	46	52	61	32–75	0.001
STAI-Y: state anxiety	41	47	52	27–67	45	51	56	30–76	0.01

Q1: 25th percentile, Q2: 50th percentile, Q3: 75th percentile. Wilcoxon signed-rank test.

**Table 3 ijerph-19-12829-t003:** Outcomes for the 1st intervention vs. the 2nd intervention.

									*n* = 83
	1st Intervention (Guidebook)				2nd Intervention (Digital Animation)				*p* Value
		(Median)				(Median)			
	Q1	Q2	Q3	Range	Q1	Q2	Q3	Range	
*Mari Meter-x*	(−2)	0	3	(−9)–24	(−1)	2	9	(−14)–30	0.005
STAI-Y: state anxiety	(−1)	2	3	(−15)–14	0	2	6	(−10)–20	0.088

Q1: 25th percentile, Q2: 50th percentile, Q3: 75th percentile. Wilcoxon rank sum test.

**Table 4 ijerph-19-12829-t004:** Outcomes for the 1st intervention vs. the 1st intervention & 2nd intervention.

									*n* = 83
	1st Intervention (Guidebook)				1st Intervention (Guidebook) & 2nd Intervention (Digital Animation)				*p* Value
		**(Median)**				**(Median)**			
	**Q1**	**Q2**	**Q3**	**Range**	**Q1**	**Q2**	**Q3**	**Range**	
*Mari Meter-x*	(−2)	0	3	(−9)–24	(−1)	7	13	(23)–29	0.001
STAI-Y: state anxiety	(−1)	2	3	(−15)–14	(−10)	(−2)	5	(−29)–17	0.003

Q1: 25th percentile, Q2: 50th percentile, Q3: 75th percentile. Wilcoxon rank sum test.

## Data Availability

The datasets used during the current study are available from the corresponding author upon reasonable request.

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
