# Peer review of "Cross-Cultural Information for Japanese Nurses at an International Hospital: A Controlled Before–After Intervention Study"

_ijerph, 2022, doi:10.3390/ijerph191912829_

Round 1

Reviewer 1 Report

The data was collected in 2016, which is considered too old in my opinion. And the nursing implications of interventions are rarely found.

Author Response

Response for reviewer 1

Comments and Suggestions for Authors

Reviewer: The data was collected in 2016, which is considered too old in my opinion. And the nursing implications of interventions are rarely found.

Author: We believe that it is valuable for nurses to gain educational knowledge about the visitors in an easy-to-understand video. Such opportunities are rare and the nurses who watched the video felt positive that they were able to figure out things they did not know. The nursing situation will not change much in 2022 compared to 2016. Few studies have been done on this topic. For example:

  1. Kaihlanen AM, Hietapakka L, Heponiemi T., Increasing cultural awareness: qualitative study of nurses' perceptions about cultural competence training. BMC Nurs 2019, 18:38.
  2. Benjamin GS. erception and Experience of Transcultural Care of Stakeholders and Health Service Users with a Migrant Background: A Qualitative Study. Int J Environ Res Public Health. 2021, 7;18(19).

Reviewer 2 Report

The results of this study are interesting. However, please consider the following points and revise your manuscript. I hope my comments will be helpful to your manuscript.

Line 32-33

It seemed incongruous that COVID-19 in 2019 would be mentioned in the introduction when the survey period was 2016.

 Line 152: 2.2.1. Sample size

Please describe what software was used and how the sample size was calculated. The appropriateness of the sample size should be described in Section 2.4.3, Data analysis. In doing so, it is more consistent with the statistical processing method.

 Line 155: 2.2.2. Eligibility

How did you assess the English proficiency of the subjects in this study to "read and understand English"? Please describe the method used.

Line 232: In the 3.2. Reliability of outcomes

Authors mentioned that "We calculated the Cronbach's alpha internal consistency reliability coefficient to assess the reliability of the measures used in this study.” However, this is not stated in the statistical method section. The authors should explain the analysis method in the statistical methods section.

  Also, how did you set the significance level for the overall statistical treatment method? Please describe it in the statistical methods section.

 About the tables:

Table 1 contains abbreviations such as ICU/Ope/ER. Please add an explanation of the abbreviations under the table. 

In Tables 2, 3, and 4, some words are broken in the middle of the word. Please correct them.

Please be consistent with either listing the p-value or listing the asterisk. If p-values are listed in the table, there is no need to include an asterisk in the table.

 Line 239

The statistical processing method described in the analysis method is described several times in the results. It would be better to remove the duplicate wording.

 Line 240-251

Following expression is more appropriate:

 In the first intervention group, the Mari Meter-X score did not change significantly (p=0.151) after the intervention, whereas the STAI-Y state anxiety score changed significantly (p=0.001).

A statistically significant reduction in the Mari Meter-X score from before to after the intervention was observed in the second intervention group (p=0.001). Similarly, the reduction in the STAI-Y state anxiety score from before to after receiving the second intervention was statistically significant (p=0.001). A statistically significant reduction in the Mari Meter-X score from before to after treatment was observed in the combined first and second intervention groups (p=0.001). However, there was a statistically significant increase in 249 the STAI-Y state anxiety scores from before to after receiving the first and second intervention (p=0.01).

 Line 254-257

The following expression is more appropriate:

The Mari Meter-X score was significantly reduced in the second intervention group compared with that in the first intervention group (p<0.005). The STAI-Y state of anxiety was not significantly changed in the first intervention group and the second intervention group (p=0.088).

 Line 254-257

The following expression is more appropriate:

The Mari Meter-X score was significantly reduced in the second intervention group compared with that in the first intervention group (p<0.005). The STAI-Y state of anxiety was not significantly changed in the first intervention group and the second intervention group (p=0.088).

 Line 257-258

The following text is an interpretation of the results. This sentence should be moved to the discussion.

There were no factors included in this study that affected the outcome because it was a controlled before-after intervention study with the same nurses.”

 Line 262-266

The following expression is more appropriate:

 The Mari Meter-X score was significantly decreased in the combined first and second intervention groups compared with that in the first intervention group (p=0.001). However, the STAI-Y state anxiety scores were significantly different in the first intervention group and the first and second intervention groups (p=0.003).

 Line 266-268

The following sentences are an interpretation of the results. This sentence should be moved to the discussion.

 One factor that may have affected the outcome was that the head nurse suddenly changed between the interventions. This may not be common in hospitals. However, it is unclear whether this affected the outcome.”

Author Response

Response for reviewer 2

The results of this study are interesting. However, please consider the following points and revise your manuscript. I hope my comments will be helpful to your manuscript.

Reviewer: Are the conclusions supported by the results? Must be improved.

Author: We rewrote the conclusions (5. Conclusions) in the main text section to address the findings from this study.

Reviewer: Line 32-33

It seemed incongruous that COVID-19 in 2019 would be mentioned in the introduction when the survey period was 2016.

Author: We have excluded this sentence.

Reviewer: Line 152: 2.2.1. Sample size

Please describe what software was used and how the sample size was calculated. The appropriateness of the sample size should be described in Section 2.4.3, Data analysis. In doing so, it is more consistent with the statistical processing method.

Author: We used the JMP statistical package (version 14.0) for all statistical evaluations. We moved the description for the sample size to Section 2.4.3 (Data analysis).

Line 216-222.

To achieve confidence levels of 95% and 80% and detect a difference of 0.2, we deter-mined that a sample size of 80 nurses was necessary for each intervention. The ration-al was based on previous research examining the effects of video guidance on anxiety reduction [25]. We calculated the Cronbach’s alpha internal consistency reliability coefficient to assess the reliability of the measures used in this study. The Mari Meter-X yielded a reliability of 0.88, while the STAI-Y yielded a reliability of 0.91. All statistical evaluations were performed with the help of the JMP statistical package (version 14.0).

Reviewer: Line 155: 2.2.2. Eligibility

How did you assess the English proficiency of the subjects in this study to "read and understand English"? Please describe the method used.

Author: We added the description regarding the evaluation of the English proficiency in Section 2.2.1 (Eligibility) as follows:

“These nurses' assessment of their ability to read and understand English was determined by the head nurse and the nurses themselves.”

Reviewer: Line 232: In the 3.2. Reliability of outcomes

Authors mentioned that "We calculated the Cronbach's alpha internal consistency reliability coefficient to assess the reliability of the measures used in this study.” However, this is not stated in the statistical method section. The authors should explain the analysis method in the statistical methods section.

Author: We modified the explanations on the internal consistency in Section 2.4.3 (Data analysis).

Reviewer:  Also, how did you set the significance level for the overall statistical treatment method? Please describe it in the statistical methods section.

 Author: We added the description on the significance level in Section 2.4.3 (Data analysis) as follows:

“To achieve confidence levels of 95% and 80% and detect a difference of 0.2, we determined that a sample size of 80 nurses was necessary for each intervention. The rational was based on previous research examining the effects of video guidance on anxiety reduction [25].”

Reviewer: About the tables: Table 1 contains abbreviations such as ICU/Ope/ER. Please add an explanation of the abbreviations under the table. 

Author: We added the explanation of these terms below Table 1.

Reviewer: In Tables 2, 3, and 4, some words are broken in the middle of the word. Please correct them.

Author: We reviewed the row and column headings in Tables 2, 3, and 4 and corrected the words.

Reviewer: Please be consistent with either listing the p-value or listing the asterisk. If p-values are listed in the table, there is no need to include an asterisk in the table.

Author: We amended tables by deleting asterisks and showing p-values only.

Reviewer: Line 239

The statistical processing method described in the analysis method is described several times in the results. It would be better to remove the duplicate wording.

Author: We removed the descriptions for the names of statistical analysis methods from the Results section.

 Reviewer: Line 240-251

Following expression is more appropriate:

 In the first intervention group, the Mari Meter-X score did not change significantly (p=0.151) after the intervention, whereas the STAI-Y state anxiety score changed significantly (p=0.001).

Author: We revised the sentence accordingly. (Lines 243-)

Reviewer: A statistically significant reduction in the Mari Meter-X score from before to after the intervention was observed in the second intervention group (p=0.001). Similarly, the reduction in the STAI-Y state anxiety score from before to after receiving the second intervention was statistically significant (p=0.001). A statistically significant reduction in the Mari Meter-X score from before to after treatment was observed in the combined first and second intervention groups (p=0.001). However, there was a statistically significant increase in 249 the STAI-Y state anxiety scores from before to after receiving the first and second intervention (p=0.01).

Author: We revised the sentences accordingly. (Lines 246-)

 Reviewer: Line 254-257

The following expression is more appropriate:

The Mari Meter-X score was significantly reduced in the second intervention group compared with that in the first intervention group (p<0.005). The STAI-Y state of anxiety was not significantly changed in the first intervention group and the second intervention group (p=0.088).

Author: We revised the sentences accordingly. (Lines 255-)

Reviewer: Line 257-258

The following text is an interpretation of the results. This sentence should be moved to the discussion.

“There were no factors included in this study that affected the outcome because it was a controlled before-after intervention study with the same nurses.”

Author: We have moved the sentence to the Discussion section. (Lines 315-).

 Reviewer: Line 262-266

The following expression is more appropriate:

 The Mari Meter-X score was significantly decreased in the combined first and second intervention groups compared with that in the first intervention group (p=0.001). However, the STAI-Y state anxiety scores were significantly different in the first intervention group and the first and second intervention groups (p=0.003).

Author: We revised the sentences accordingly. (Lines 260-)

 Reviewer: Line 266-268

The following sentences are an interpretation of the results. This sentence should be moved to the discussion.

 “One factor that may have affected the outcome was that the head nurse suddenly changed between the interventions. This may not be common in hospitals. However, it is unclear whether this affected the outcome.”

Author: We removed these sentences.

Reviewer 3 Report

The article presented for review concerns a very important issue related to the comfort of nurses' work. One of the important aspects of the work of nurses is communication with the patient. Its high quality is a prerequisite for an effective treatment process. One of the important and significant barriers is cultural differences. This applies - as the authors mention - to the work of Japanese nurses in contact with foreigners.

The structure of the article is correct, the tests used are correct, the statistical analysis and the presentation of the results is very good.

I only have two little comments: In the first part of the article  some information on the importance of visual media in improving communication could be added, it seems to be a very significant direction nowadays.

The description of the video material could be more detailed - which is mainly related to the information gathered in 11 points?

Author Response

Response for reviewer 3

Reviewer: Does the introduction provide sufficient background and include all relevant references?  Can be improved

Author: We reviewed the Introduction section and added a paragraph to explain previous studies that implied the importance of a culturally relevant, easy-to-watch video related to healthcare in Japan. (Lines 80-).

Comments and Suggestions for Authors

Reviewer: The article presented for review concerns a very important issue related to the comfort of nurses' work. One of the important aspects of the work of nurses is communication with the patient. Its high quality is a prerequisite for an effective treatment process. One of the important and significant barriers is cultural differences. This applies - as the authors mention - to the work of Japanese nurses in contact with foreigners.

The structure of the article is correct, the tests used are correct, the statistical analysis and the presentation of the results is very good.

I only have two little comments: In the first part of the article some information on the importance of visual media in improving communication could be added, it seems to be a very significant direction nowadays.

Author: We found some up-to-date videos on this topic; however, we decided not to reflect them on this manuscript as we focused on the cultural context in the healthcare provision for foreign patients, which was not seen in these videos.

Reviewer: The description of the video material could be more detailed - which is mainly related to the information gathered in 11 points?

Author: We added the following sentences with references in the introduction section: line 81

“We conducted a cross-sectional survey of nurses and foreign visitors to Japan prior to this study because we could not find a culturally relevant, easy-to-watch health-related video in Japan. We analyzed the results and produced a video named Mari Info Japan with information on 11 items needed for health-related cultural background education [19,25].”

Round 2

Reviewer 1 Report

The dependent variable in this study was anxiety (measured by STAI-Y). In the introduction, why anxiety should be viewed as a dependent variable and how the intervention affects the lowering of anxiety should be further emphasized and described.
